# Assessment of the Impact of the Addition of Nanoparticles on the Properties of Glass–Ionomer Cements

**DOI:** 10.3390/ma13020276

**Published:** 2020-01-08

**Authors:** Elizabeta Gjorgievska, John W. Nicholson, Dragana Gabrić, Zeynep Asli Guclu, Ivana Miletić, Nichola J. Coleman

**Affiliations:** 1Department of Paediatric and Preventive Dentistry, Faculty of Dentistry, University of Skopje, 1000 Skopje, Macedonia; elizabetag2000@yahoo.com; 2Bluefield Center of Biomaterials, London EC1N 8JY, UK; john.nicholson@bluefieldcentre.co.uk; 3Department of Oral Surgery, School of Dental Medicine, University of Zagreb, 10000 Zagreb, Croatia; 4Faculty of Dentistry, University of Kayseri, 38010 Kayseri, Turkey; zaguclu@gmail.com; 5Department of Endodontics and Restorative Dentistry, School of Dental Medicine, University of Zagreb, 10000 Zagreb, Croatia; miletic@sfzg.hr; 6School of Science, University of Greenwich, Central Ave, Gillingham, Chatham ME4 4TB, UK; n.coleman@gre.ac.uk

**Keywords:** glass-ionomer cements, nanoparticles, titanium dioxide, zirconium oxide, aluminium oxide

## Abstract

The aim of the study was to evaluate the effects of incorporation of Al_2_O_3_, ZrO_2_ and TiO_2_ nanoparticles into glass–ionomer cements (GICs). Two different GICs were used in the study. Four groups were prepared for each material: the control group (without nanoparticles) and three groups modified by the incorporation of nanoparticles at 2, 5 or 10 wt %, respectively. Cements were mixed and placed in moulds (4 mm × 6 mm); after setting, the samples were stored in saline (one day and one week). Compressive strengths were measured and the morphology of the fractured surfaces was analyzed by scanning electron microscopy. The elements released into the storage solutions were determined by Inductively coupled plasma-optical emission spectrometry (ICP-OES). Addition of nanoparticles was found to alter the appearance of cements as examined by scanning electron microscopy. Compressive strength increased with the addition of ZrO_2_ and especially TiO_2_ nanoparticles, whereas the addition of Al_2_O_3_ nanoparticles generally weakened the cements. The ion release profile of the modified cements was the same in all cases. The addition of Al_2_O_3_, ZrO_2_ and TiO_2_ nanoparticles into GICs is beneficial, since it leads to reduction of the microscopic voids in the set cement. Of these, the use of ZrO_2_ and TiO_2_ nanoparticles also led to increased compressive strength. Nanoparticles did not release detectable levels of ions (Al, Zr or Ti), which makes them suitable for clinical use.

## 1. Introduction

Glass ionomer (polyalkenoate) cements (GICs) are widely used dental restorative materials, especially in paediatric dentistry. Based on the principle of “biomimesis” (i.e., replacement of a natural tissue using artificial materials that closely replicate the original structure and/or function), their properties make them suitable for use as dentin replacement materials [1,2,3,4,5].

Glass–ionomer cements consist of calcium or strontium alumino-fluoro-silicate glass powder (base) that is combined with a water-soluble acidic polymer. When these components are mixed together, they undergo an acid–base (neutralisation) setting reaction to form a hardened material [1]. Without any loss of physical properties of the hardened cement, significant amounts of fluoride ions are released from this material, a property which is crucial for their anticariogenicity, which is one of the advantages of these materials. Another major advantage of GIC is its ability to chemically bond to dentin and enamel and to form an acid resistant interface with these substrates [6]. The resulting seal is both technique tolerant and long lasting, even under challenging clinical environments. Glass–ionomers have been described as bioactive materials due to the exchange of ions with the tooth surface [7]. In addition, GIC are thermally compatible with enamel, biocompatible and of low toxicity.

However, at present, glass–ionomer cements are not perfect. They lack toughness, have low fracture strength, and also low wear resistance. These deficiencies are generally held to limit their use in the stress-bearing areas [8]. For this reason numerous modifications have been made over the years in attempts to improve the physical properties of these materials. One approach has been to employ glass particles with controlled particle sizes in such a range that high powder to liquid ratios can be achieved. This results in so-called high-viscosity glass–ionomers, and studies confirmed that these materials have superior physical properties compared with conventional GICs [9]. Other approaches have included the addition of a variety of fillers. These have included amalgam alloys and stainless steel powders, carbon and aluminosilicate fibres, and hydroxyapatite powders of various compositions [10,11,12]. Results with these additives have been variable, but some have resulted in materials with improved physical properties, notably in terms of toughness and wear resistance.

Most recently, nanoparticles such as titanium dioxide (TiO_2_) nanotubes, nanohydroxyapatite, and nanofluoroapatite have been incorporated into glass–ionomers in attempts to enhance their mechanical strength [13,14,15,16,17]. This has met with some success, and resulting cements were found to be stronger in compression than those without added nanoparticles. There is also evidence of improved biocompatibility of cements containing zirconia (ZrO_2_) nanoparticles [17]. Modified GICs produced by incorporation of different types of nanoparticles have been shown to have fewer air voids and internal microcracks, presumably because they are easier to mix than unmodified cements. In some cases this has resulted in greater compressive strengths [18].

The purpose of the present study was to evaluate the effects of incorporation three different metal oxide nanoparticles at levels of 2, 5, and 10 wt % into high-viscosity conventional GICs. Properties evaluated were mechanical strength, quality of the resulting cement matrix and ion release. The null hypotheses were that there were no differences in the microscopic appearance, compressive strengths or ion release following addition of the nanoparticles.

## 2. Materials and Methods

Two commercially available conventional GICs were used in the study: ChemFil^®^ Rock (Dentsply DeTrey, Konstanz, Germany) and GC Equia™ Fil (GC Europe N.V., Leuven, Belgium). The respective compositions of these GICs are listed in Table 1.

Four groups consisting of 12 samples were prepared for each material (a total number of 96 samples) by mixing the GICs in a capsule mixer according to the manufacturers’ instructions. The first group served as a control (without addition of nanoparticles), while the other three groups were modified by incorporation of nanoparticles, namely Al_2_O_3_, ZrO_2_ and TiO_2_ (Table 2), each at 2%, 5% and 10% by weight. The powders were characterized uncoated using a field-emission gun scanning electron microscope (FEG-SEM Hitachi SU 8030, Tokyo, Japan) and X-ray powder diffraction (XRPD, D8 Advance X-ray Diffractometer, Bruker, Karlsruhe, Germany).

The nanoparticles were mixed with the glass–ionomer by spatulation on a ceramic tile to obtain the most uniform distribution possible of the nanoparticles, after extrusion from the capsule.

The freshly mixed cement was placed in a cylindrical metal mould (4 mm in diameter, 6 mm in height) and covered at the both sides with metal slides, clamped and left in an incubator (at 37 °C) for 1 h to allow setting. Following setting, the samples were stored in physiological saline and the storage periods were 1 day and 1 week, at room temperature.

### 2.1. Compressive Strength Measurement

The compressive strength test was performed after 1 day on half of the samples from each group (6) and the rest of the samples (6) were tested after 1 week, according to the method described in ISO 9917-1:2007 [19], using a Universal Testing Machine (Instron Model 1193, Instron Corp., Canton, OH, USA) at a crosshead speed of 1 mm/min.

### 2.2. Scanning Electron Microscopy in Secondary Electron Mode (SEM-SE)

The fractured samples after the compressive strength testing were mounted onto aluminium stubs covered with conductive carbon tape. The morphologies of the fractured surfaces were analyzed uncoated in high vacuum by SEM (Quanta™ 250 Scanning Electron Microscope, FEI™ Comp. Thermo Fisher Scientific, Hillsboro, OR, USA), using the following parameters: 10 kV accelerating voltage, 5–6 mm working distance, 4 spot size, at different magnifications.

### 2.3. Inductively Coupled Plasma Analysis (ICP)

The levels of zinc (Zn), zirconium (Zr), aluminium (Al), strontium (Sr), titanium (Ti), and calcium (Ca) released from the materials into the storage solution (physiological saline) were measured after 1 week, when the materials were completely set.

The solution concentrations of the elements were monitored by inductively coupled plasma (ICP) analysis using a TJA Iris simultaneous ICP-OES spectrophotometer (SPECTRO Analytical Instruments Inc. Mahwah, NJ, USA) and multi-element standards. The procedure was undertaken in triplicates and was carried out as previously described by Hurt et al. [20].

### 2.4. Statistical Analysis

The statistical analysis of the obtained data was performed by means of descriptive analysis, multivariate analysis of variance (MANOVA) and analysis of variance (ANOVA). If statistically significant differences appeared, Post-hoc Tukey’s Honest Significant Differences (HSD) test was applied and the level of significance was *p* = 0.05.

The software used was SAS for Windows platform (SAS University Edition, SAS Cary, NC, USA).

## 3. Results

Compressive strength values varied according to the particular brand of glass–ionomer cement, the chemical composition of the nanoparticles, the loading of nanoparticles and the storage time (Figure 1). In the case of GC Equia™ Fil, all loadings of Al_2_O_3_ gave lower compressive strengths relative to the control after 1 day’s storage. This pattern changed, so that after 1 week’s storage, compressive strength values for the 2% and 10% loading were about the same as the control, and only the 5% loading gave weaker samples.

The pattern that the 5% loading was weak relative to the 2% and 10% loadings occurred in several other cases: ChemFil^®^ Rock with Al_2_O_3_ at both 1 day and 1 week storage time, GC Equia™ Fil and ChemFil^®^ with ZrO_2_ at both 1 day and 1 week storage, and GC Equia™ Fil with TiO_2_. Rock ChemFil^®^ Rock showed distinctive patterns with TiO_2_, with all loadings giving stronger samples at 1 day and 1 week, and in the latter case, the strength rising to a maximum with 10% loading. These values were not always statistically significant, though they were in most cases. There was no significant difference between the materials after 1 week.

The SEM micrographs show that the addition of the nanoparticles reduced the porosity of the glass–ionomer cements. SEM microphotographs of the control sample of GC Equia™ Fil after 1 day storage in saline showed many air voids within the matrix. There were also many fracture lines visible (Figure 2A) With the addition of 2%, 5% and 10% of Al_2_O_3_, ZrO_2_, TiO_2_, the number of voids was found to decrease, and fractures increasingly ran through the matrix. The most clear-cut example of this was with GC Equia™ Fil with 10% TiO_2_ after 1 week storage (Figure 3A,B).

Similar results were obtained with ChemFil^®^ Rock, but the results shown in the microphotographs (Figure 4A,B, and Figure 5A,B) demonstrate that the greatest reduction in porosity occurred following addition of 5% of Al_2_O_3_, ZrO_2_, TiO_2_.

The means and standard deviations for the ion release measured by ICP-OES are shown in Table 3. The release of zirconium and titanium were below detection levels in cases. The level of aluminium release increased with Al_2_O_3_ nanoparticles, especially in GC Equia™ Fil, the highest being with 5% loading.

The ANOVA test (shown in Table 4) for the concentration of the elements shows that the concentration of zinc is different between the two materials. For strontium, all factors and interactions are significant, except between the material and the concentration, while for titanium all of the factors and interactions are significant. The element and the concentration are significant for phosphorus and silicon ion release.

The MANOVA test results (Table 4) for the *p*-value for Wilks’ λ indicate that the release of ions is different between the two types of the materials, the type of the nanoparticles incorporated and their loadings. The interaction of these factors is also significant.

## 4. Discussion

Results show that the addition of nanoparticles alters the appearance of the fracture surfaces of the cements as examined by scanning electron microscopy. This change is generally associated with the improvements in the compressive strength. It is known that the pores in a solid body act as stress-concentration points where fracture can initiate [21]. Once the material sets, these voids become trapped in the cement where they cause stress to concentrate and thus become points of mechanical weakness [15]. The results of the present study showed that there is an increase in the compressive strength of the GICs following incorporation of ZrO_2_ and especially TiO_2_ nanoparticles, unlike the addition of Al_2_O_3_ nanoparticles. Particle size and particle size distribution can be seen to have substantial effects on the microstructure of the cement and hence on the mechanical properties. Prentice et al. [22] demonstrated that an increased proportion of smaller particles led to higher strengths, and that an increased proportion of larger particles corresponded with a decrease in the viscosity of the unset cement paste. The optimisation of the particle sizes and distribution can lead to GIC with improved properties and these are likely to improve the longevity of the restoration.

It seems likely that the smaller particle size of the first two types of nanoparticle is responsible for the increase of the strength, rather than their chemical composition. The TiO_2_ nanoparticles were the smallest of the nanoparticles studied, and these particles gave particularly promising results. Similar results have been reported previously, notably the increase in the strength of GICs by incorporating TiO_2_ nanoparticles [18]. TiO_2_ nanoparticles have also been proposed as reinforcing fillers for dental resin composites and epoxies [14,15,16,17].

Previous studies have provided evidence of improved biocompatibility of cements containing zirconia (ZrO_2_) nanoparticles [17]. This is despite the fact that there are concerns about the toxicity of nanoparticles that arise from the ease with they can pass through physiological barriers and lead adverse effects once inside cells. Recent studies have confirmed that these nanoparticles have selective toxicity towards bacteria [23]. However, they have minimal effects on human cells. The current study employed nanoparticles that are known to be of relatively low toxicity, but the full extent of their lack of toxicity is not clear. For example, recent studies suggest that TiO_2_ nanoparticles may possess higher toxicity potential than their bulk materials for reasons that are not fully understood [24]. Also, ZrO_2_ nanoparticles have been found to cause increases in levels of free radicals within cells, a feature which leads to damage [17]. Lastly, Al_2_O_3_ nanoparticles are known to disturb the cell viability, alter the mitochondrial function, increase the oxidative stress, alter tight junction protein expression of the blood brain barrier and damage DNA within cells [25].

It is important that the dental materials used are safe for use in patients. In the current study, the ion release profile of the modified cements was studied, in order to verify that the nanoparticles employed do not leach out of the cements. For all three types of nanoparticle, no detectable change in ion release was observed. This suggests that these materials are safe for use in the mouth.

## 5. Conclusions

The addition of Al_2_O_3_, ZrO_2_ and TiO_2_ nanoparticles into high-viscosity conventional GICs leads to a reduction in the presence of microscopic voids in the set cement, a feature which was frequently found to be associated with an increase in compressive strength. No ion leaching from the nanoparticles could be detected. Overall, we conclude that the addition of these nanoparticles is beneficial for glass–ionomer cements and has the potential to lead to substantial improvements in properties in materials for use in human subjects under clinical conditions.

## Figures and Tables

**Figure 1 materials-13-00276-f001:**
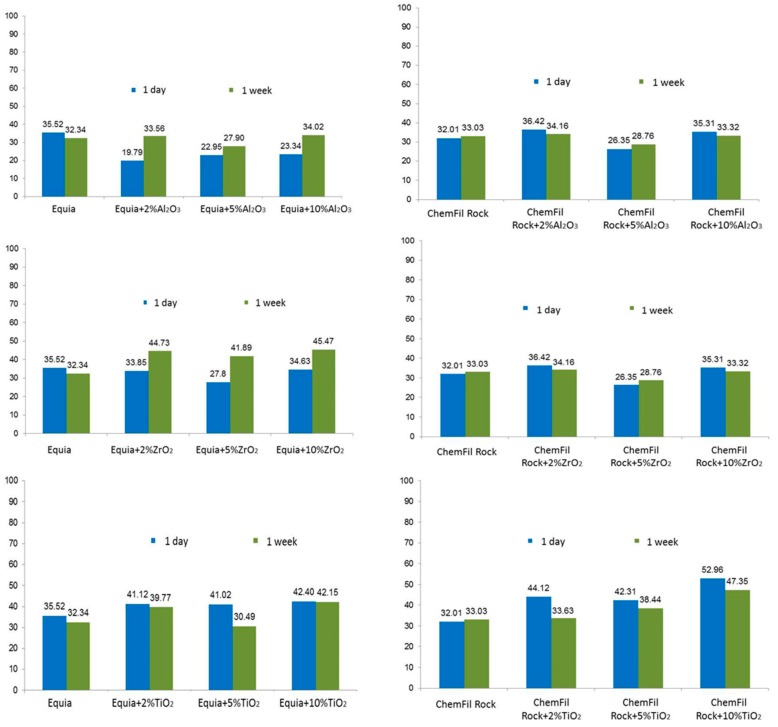
Compressive strength of the GICs without and with addition of Al_2_O_3_, ZrO_2_ and TiO_2_ nanoparticles after 1 day and 1 week of storage into physiological saline.

**Figure 2 materials-13-00276-f002:**
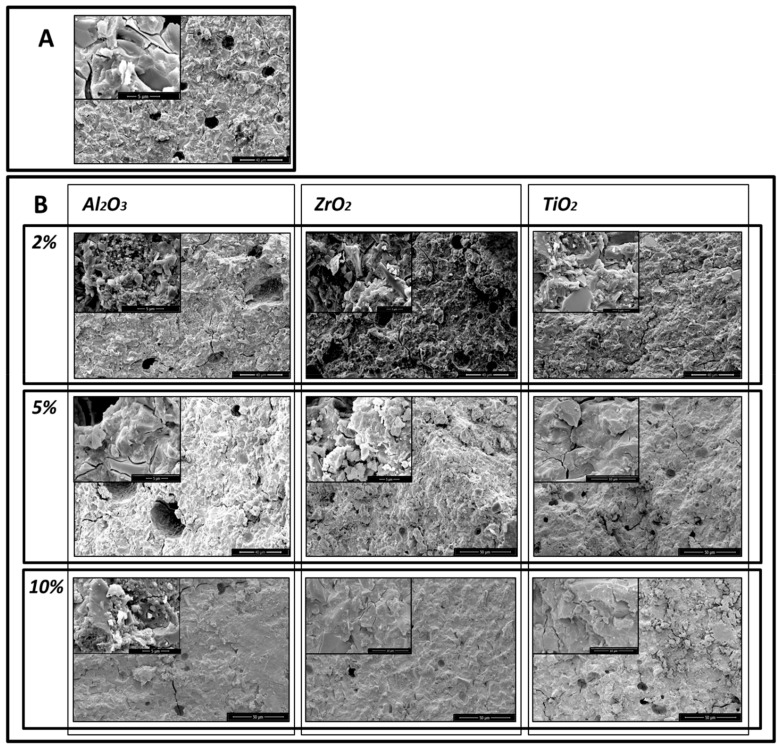
Microphotographs of GC EquiaFil after 1day storage: (**A**) the original materials; (**B**) after addition of 2%, 5% and 10% of Al_2_O_3_, ZrO_2_, TiO_2_.

**Figure 3 materials-13-00276-f003:**
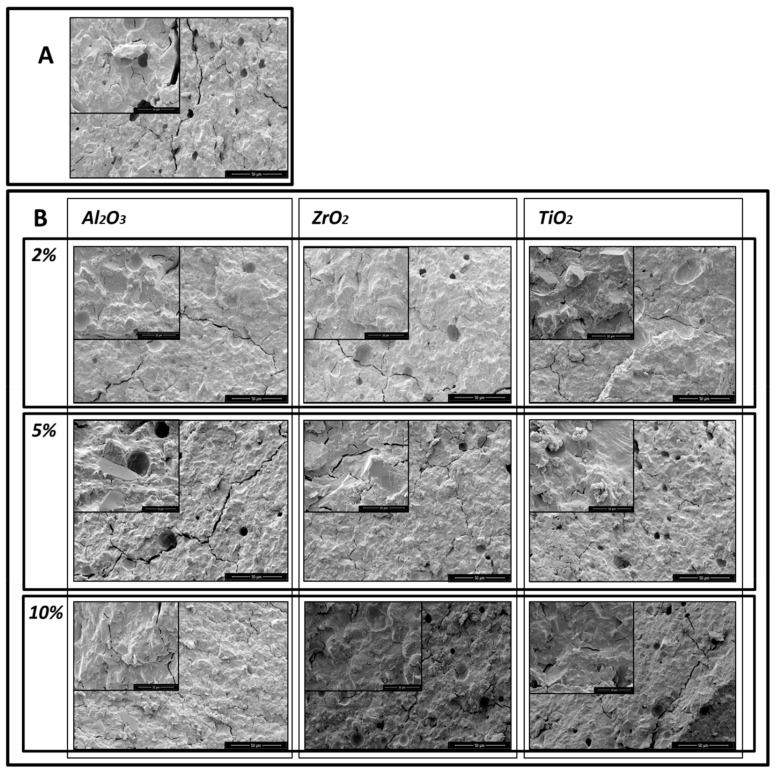
Microphotographs of GC EquiaFil after 1 week storage: (**A**) the original materials; (**B**) after addition of 2%, 5% and 10% of Al_2_O_3_, ZrO_2_, TiO_2_.

**Figure 4 materials-13-00276-f004:**
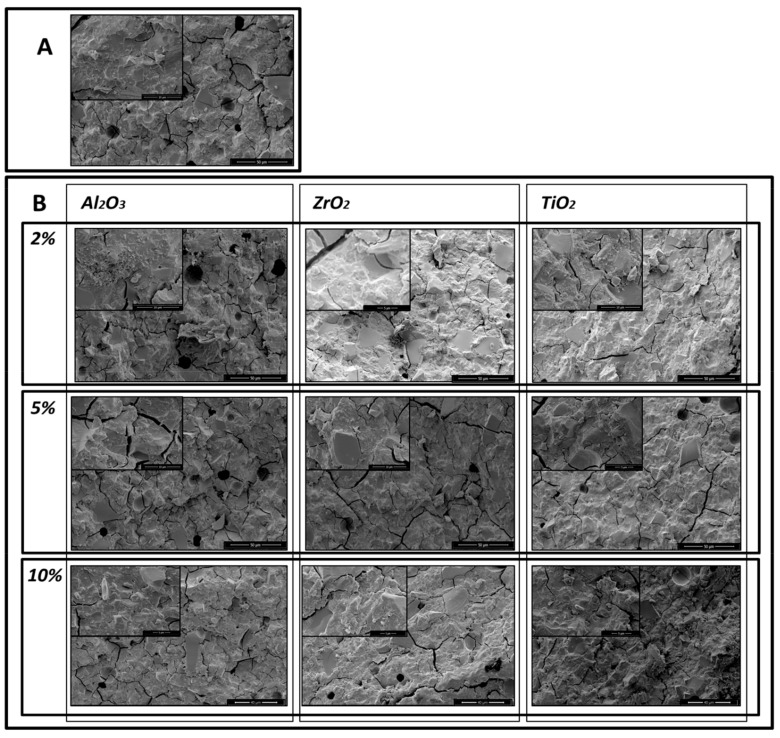
Microphotographs of ChemFil Rock after 1 day storage: (**A**) the original materials, without modification; (**B**) after addition of 2%, 5% and 10% of Al_2_O_3_, ZrO_2_, TiO_2_.

**Figure 5 materials-13-00276-f005:**
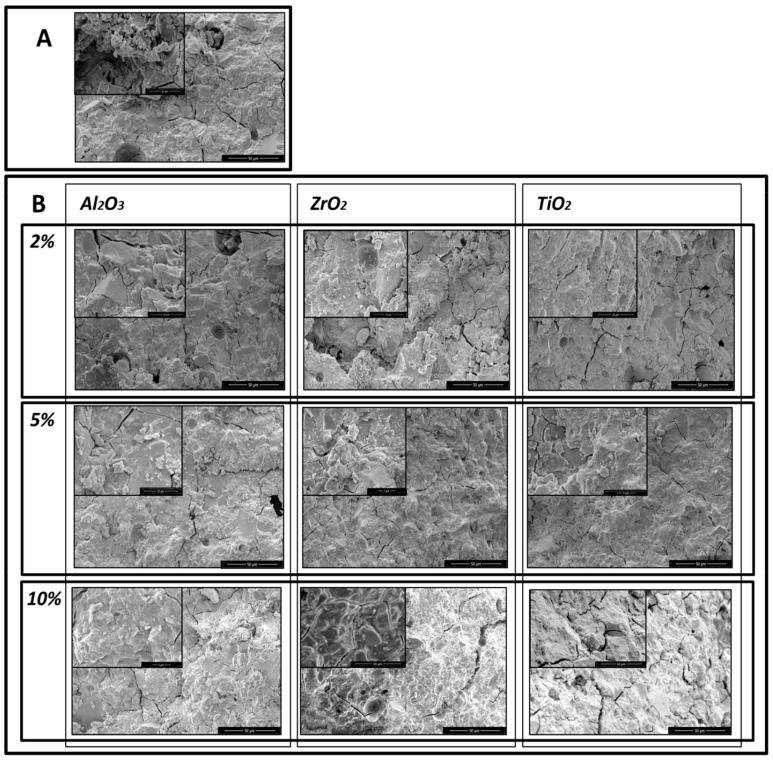
Microphotographs of ChemFil Rock after 1 week storage: (**A**) the original materials; (**B**) after addition of 2%, 5% and 10% of Al_2_O_3_, ZrO_2_, TiO_2_.

**Table 1 materials-13-00276-t001:** Restoratives used in the study.

Material	Classification	Composition	Manufacturer
GC EQUIA™ Fil	Conventional glass–ionomer cement	Polyacrylic acid, aluminosilicate glass, distilled water	GC Europe N.V., Leuven, Belgium
ChemFil^®^ Rock	Conventional glass–ionomer cement	Calcium-aluminium-zinc-fluoro-phosphor-silicate glass, polycarboxylic acid, iron oxide pigments, titanium dioxide pigments, tartaric acid, water	DENTSPLY DeTrey, Konstanz, Germany

**Table 2 materials-13-00276-t002:** Nanoparticles incorporated into the glass–ionomer cement (GIC, as certified by the manufacturer).

Name	Formula	Dimensions	Manufacturer
Zirconia	ZrO_2_	80 nm	Nanoshel, Punjab, India
Titania	TiO_2_	10–25 nm	Nanoshel, Punjab, India
Alumina	Al_2_O_3_	<100 nm	Nanoshel, Punjab, India

**Table 3 materials-13-00276-t003:** Ion release into physiological saline, measured by inductively coupled plasma (ICP), after 1 day following storage of GICs without and with addition of Al_2_O_3_, ZrO_2_ and TiO_2_ nanoparticles.

Material	Ion	Control	Zr			Al			Ti		
			2%	5%	10%	2%	5%	10%	2%	5%	10%
**Equia Fil**	Zn	0.00 (0.00)	0.00 (0.00)	0.00 (0.00)	0.00 (0.00)	0.00 (0.00)	0.00 (0.00)	0.00 (0.00)	0.00 (0.00)	0.00 (0.00)	0.00 (0.00)
	Zr	0.00 (0.00)	0.00 (0.00)	0.00 (0.00)	0.00 (0.00)	0.00 (0.00)	0.00 (0.00)	0.00 (0.00)	0.00 (0.00)	0.00 (0.00)	0.00 (0.00)
	Al	7.49 (0.94)	6.71 (0.95)	7.88 (1.56)	7.03 (1.81)	8.57 (1.20)	12.41 (2.67)	10.27 (1.68)	6.25 (1.42)	10.52 (0.71)	9.71 (3.80)
	Sr	0.28 (0.18)	0.81 (0.53)	0.55 (0.22)	0.53 (0.11)	1.16 (0.46)	1.62 (1.16)	1.04 (0.63)	0.83 (0.59)	3.98 (1.14)	0.95 (0.21)
	Ti	0.00 (0.00)	0.00 (0.00)	0.00 (0.00)	0.00 (0.00)	0.00 (0.00)	0.00 (0.00)	0.00 (0.00)	0.00 (0.00)	0.01 (0.01)	0.00 (0.00)
	Ca	0.06 (0.08)	0.08 (0.17)	0.00 (0.00)	0.00 (0.00)	0.00 (0.00)	0.09 (0.10)	0.02 (0.06)	0.03 (0.05)	0.65 (0.59)	0.07 (0.10)
	P	0.62 (0.08)	0.81 (0.19)	0.82 (0.08)	0.73 (0.36)	1.60 (1.26)	3.18 (2.17)	1.58 (0.98)	1.12 (0.62)	5.52 (2.07)	1.16 (0.80)
	Si	2.82 (0.29)	2.76 (0.48)	2.99 (0.67)	2.81 (0.94)	4.76 (1.40)	7.95 (4.03)	4.48 (2.03)	3.14 (1.52)	13.90 (4.51)	4.27 (2.70)
**Chem Fil**	Zn	0.94 (0.09)	0.90 (0.23)	1.45 (0.48)	1.09 (0.59)	1.19 (0.40)	1.06 (0.41)	1.21 (0.25)	1.13 (0.45)	1.61 (0.56)	1.19 (0.55)
	Zr	0.00 (0.00)	0.00 (0.00)	0.00 (0.00)	0.00 (0.00)	0.00 (0.00)	0.00 (0.00)	0.00 (0.00)	0.00 (0.00)	0.00 (0.00)	0.00 (0.00)
	Al	0.23 (0.07)	0.62 (0.18)	0.81 (0.28)	0.73 (0.47)	0.75 (0.29)	0.71 (0.35)	1.19 (0.82)	0.50 (0.29)	0.73 (0.34)	0.92 (0.59)
	Sr	2.33 (0.22)	2.25 (0.34)	3.29 (0.57)	2.34 (0.64)	2.50 (0.33)	2.60 (0.48)	2.82 (0.44)	2.51 (0.72)	3.00 (0.78)	2.52 (0.77)
	Ti	0.00 (0.00)	0.00 (0.00)	0.00 (0.00)	0.00 (0.00)	0.00 (0.00)	0.00 (0.00)	0.00 (0.00)	0.00 (0.00)	0.00 (0.00)	0.00 (0.00)
	Ca	0.96 (0.18)	1.05 (0.30)	1.40 (0.47)	0.84 (0.52)	1.48 (0.28)	1.75 (0.44)	1.20 (0.14)	1.16 (0.20)	1.72 (1.15)	1.51 (0.51)
	P	0.41 (0.11)	0.62 (0.28)	1.05 (0.51)	0.66 (0.19)	1.96 (0.86)	2.11 (0.97)	1.67 (0.37)	1.20 (0.89)	2.07 (1.19)	2.62 (1.53)
	Si	2.45 (0.64)	3.42 (1.39)	4.37 (1.65)	3.00 (0.74)	5.12 (1.35)	5.48 (1.35)	4.73 (0.98)	4.24 (2.06)	6.41 (3.32)	6.81 (2.22)

**Table 4 materials-13-00276-t004:** Results of MANOVA and ANOVA of the ICP analysis.

Factor	Zn	Al	Sr	Ti	Ca	P	Si	Wilks’ λ
	*p* *	*p* *	*p* *	*p* *	*p* *	*p* *	*p* *	*p*
material	0.0000	0.0000	0.0000	0.0264	0.0000	0.3676	0.7777	<0.0001
element	0.5789	0.0001	0.0001	0.0037	0.0687	0.0000	0.0000	<0.0001
material * element	0.4072	0.0054	0.0084	0.0100	0.3155	0.8996	0.5012	<0.0001
concentration	0.1749	0.0001	0.0000	0.0004	0.0038	0.0000	0.0000	<0.0001
material * concentration	0.1285	0.0014	0.1907	0.0012	0.4128	0.0035	0.0035	<0.0001
element * concentration	0.6572	0.3061	0.0002	0.0000	0.3017	0.0042	0.0005	<0.0001
material * element * concentration	0.5159	0.3304	0.0000	0.0000	0.3131	0.0026	0.0016	<0.0001

* *p*-value for ANOVA test.

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
