# Peer review of "Assessment of the Impact of the Addition of Nanoparticles on the Properties of Glass–Ionomer Cements"

_materials, 2020, doi:10.3390/ma13020276_

Round 1
Reviewer 1 Report
This paper presents findings from an experimental program that was carried to investigate the effect of adding nanoparticles to improve properties of glass-ionomer cements often used as a dental restorative material. The following items are to be addressed before this work is publishable:
Were any durability/toxicity tests carried out? How does the addition of nanoparticles affect overall cost and fabrication process of glass-ionomer cements? This study seems to be mainly concerned with compressive strength property. What about other properties? 3 contains a number of figures and illustrations. However, these are not properly discussed nor articulated.Author Response
Dear reviewer,
I would like to thank you for the useful comments and your efforts in improving the manuscript. I am addressing the issues, as follows:
1. Were any durability/toxicity tests carried out?
The testing of the ion release is in context of testing the toxicity, since the question of the toxicity of nanoparticles arises from the ease they can pass through physiological barriers and lead adverse effects once inside cells. we used relatively low toxic nanoparticles, and also, the results of the ion release show that most of the nanoparticles leached are beyond detection level. Additionally, we have performed MC 3T3 E1 cell viability test, but it is impossible to put all of the results in a single paper, therefore we decided that ion release is a sufficient test for a preliminary report such as this one.
2. How does the addition of nanoparticles affect overall cost and fabrication process of glass-ionomer cements?
As previously mentioned, this is a preliminary study to show the positive effects we found in these composite materials, but eventually it wouldn't have a drastic impact on the price, since the price of the nanoparticles is low and also, it would improve the longevity of the restorations, so in long-term this would be cost-effective.
3. This study seems to be mainly concerned with compressive strength property. What about other properties?
The study evaluated the compressive strength, but also the morphology of the materials (compactness, fracture lines) and ion release profile, as well.
4. 3 contains a number of figures and illustrations. However, these are not properly discussed nor articulated.
We have altered the Results section by adding explanation of the microphotographs (Figure 1,2,34). Please find it marked yellow in the attached document.
Reviewer 2 Report
The topic taken up by the authors is important from both a scientific and an application point of view. The research results presented in the article may complement the data on GICs composites. It is valuable that the authors used statistical methods to assess the differences between individual composites. Health safety and lack of toxicity are very important here. If such materials are to find practical application, they should also show adequate strength for a fairly long period of time. In this case, are tests carried out after 1 day and one week sufficient? In addition, in many cases, among the presented test results, one may observe a decrease in strength after 1 week compared to that tested after 1 day. Is this trend correct?
From the editing side, I suggest changing the title so that it is not an interrogative sentence, e.g. Assessment of the impact of the addition of nanoparticles on the properties of glass-ionomer cements. In my opinion, there is no need to introduce subchapter 3.2. Figures and Tables. The article would be more readable if tables and drawings appeared in the text while referring to them.
Author Response
Dear Reviewer,
Thank you for your review, and for the useful comments in order to improve the quality of the mnuscript.
It is valuable that the authors used statistical methods to assess the differences between individual composites.
We performed the statistical analysis of the obtained data by means of descriptive analysis, multivariate analysis of variance (MANOVA) and analysis of variance (ANOVA). If statistically significant differences appeared, Post-hoc Tukey’s Honest Significant Differences (HSD) test was applied and the level of significance was p=0.05.
The software used was SAS for Windows platform.
Health safety and lack of toxicity are very important here.Ion release profile was performed because of the fact that there are concerns about the toxicity of nanoparticles that arise from the ease with they can pass through physiological barriers and lead adverse effects once inside cells. The results show that there is low release, and in some cases, undetectable level of ion release from the materials with the addition of the nanopraticles. If such materials are to find practical application, they should also show adequate strength for a fairly long period of time. In this case, are tests carried out after 1 day and one week sufficient? In addition, in many cases, among the presented test results, one may observe a decrease in strength after 1 week compared to that tested after 1 day. Is this trend correct?The compressive strength showed significant increase in all the tested materias after 1 day; and a slight decrease in 5% TiO2 in EquiaFil, and decrease after addition of TiO2 in ChemFil Rock afteer 1 week. Still, the values of the compressive strength are higher than the original material. Since the GICs show increase in strength in time, we found unnecessary to do a longer-term test. From the editing side, I suggest changing the title so that it is not an interrogative sentence, e.g. Assessment of the impact of the addition of nanoparticles on the properties of glass-ionomer cements.
We have changed the title as suggested (Please see attachment) In my opinion, there is no need to introduce subchapter 3.2. Figures and Tables. The article would be more readable if tables and drawings appeared in the text while referring to them.We made the changes as suggested (Please see attachment)